# Evidence for compositionally distinct upper mantle plumelets since the early history of the Tristan-Gough hotspot

Stephan Homrighausen [1] ✉, Kaj Hoernle [1,2], Folkmar Hauff[1], Patrick A. Hoyer [3], Karsten M. Haase[3], Wolfram H. Geissler [4] & Jörg Geldmacher[1]

Recent studies indicate that mantle plumes, which transfer material and heat from the earth's interior to its surface, represent multifaceted upwellings. The Tristan-Gough hotspot track (South Atlantic), which formed above a mantle plume, documents spatial geochemical zonation in two distinct sub-tracks since ~70 Ma. The origin and the sudden appearance of two distinct geochemical flavors is enigmatic, but could provide insights into the structural evolution of mantle plumes. Sr–Nd–Pb–Hf isotope data from the Late Cretaceous Rio Grande Rise and adjacent Jean Charcot Seamount Chain (South American Plate), which represent the counterpart of the older Tristan-Gough volcanic track (African Plate), extends the bilateral-zonation to ~100 Ma. Our results support recent numerical models, demonstrating that mantle plumes can split into distinct upper mantle conduits, and provide evidence that these plumelets formed at the plume head-to-plume tail transition. We attribute the plume zonation to sampling the geochemically-graded margin of the African Large Low-Shear-Velocity Province.

The mantle plume model has been debated for decades, but a number of independent approaches including experimental, geochemical, geochronological, geophysical and numerical studies provide increasing evidence that mantle plumes are a fundamental feature of the Earth[1]. The initial hypothesis stated that linear age-progressive intraplate volcanic chains form as the plate moves over a fixed hotspot in the earth's mantle[2], later referred to as mantle plumes (pipe-like upwellings; Fig. 1), derived from the lower mantle[3]. Since some hotspot tracks initiated with the emplacement of a Large Igneous Province (LIP), it was proposed that initial mantle plumes have a large spherical plume head attached to a small conduit or plume tail.[4]. Melting of plume heads, flattened against the base of the lithosphere, produce volcanism over large areas (up to ~2000 km in diameter) over a relatively short time period of several million years (plume head stage). Once the plume head dissipates, volcanism takes place over the narrow plume tail ( ~100-300 km diameter) forming an age-progressive hotspot track on the moving lithosphere (plume tail stage). Over the past fifty years, the concept of mantle plume structure has changed significantly (Fig. 1). Numerical and seismic-tomographic studies suggest that mantle plumes have a broad dome-like rather than thin conduit-like shape below depths of ~1000 km[5], which feed single[6] or multiple thin conduits or plumelets in the upper mantle[7], which can form broad tree-like structures[8].

In addition to the variable shapes of the rising mantle plumes (Fig. 1), the internal chemical structure of plumes are also controversially described as unzoned or laterally/concentrically zoned (e.g.[9–13]). Hotspot tracks, if spatially zoned in geochemistry, can provide evidence for the internal arrangement of distinct mantle components in the mantle upwellings (plumes) and possibly their sources (e.g.[10–16]). The Tristan-Gough volcanic track represents a classic

[1]GEOMAR Helmholtz Centre for Ocean Research Kiel, Kiel, Germany. [2]Kiel University, Institute of Geosciences, Kiel, Germany. [3]Friedrich-Alexander-Universität Erlangen-Nürnberg, GeoZentrum Nordbayern, Erlangen, Germany. [4]Alfred Wegener Institute, Helmholtz Centre for Polar and Marine Research, Bremerhaven, Germany. ✉e-mail: shomrighausen@geomar.de

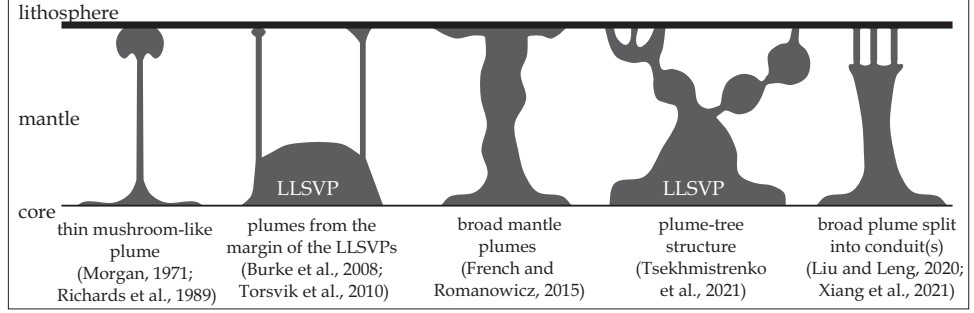

**Fig. 1 | Schematic cartoon of the proposed mantle plume structures over the past decades.** LLSVP Large Low-Shear Velocity Province; Literature data[3–8,66–68].

**Fig. 2 | Bathymetric maps of the South Atlantic Ocean showing the complex Tristan-Gough hotspot track. a** The overview map shows color-coded sample locations of the Tristan-Gough seamount chain, resolved Tristan (T) and proposed Gough (G) hotspot locations[18,22]. Ar–Ar ages from the RGR and Walvis Ridge[19,21,22]. The possible extent of the recently proposed microplate (red dashed line)[24,26], and fracture zones (thin white lines)[69]. Box in **a** shows the location of blowup of Rio Grande Rise (RGR) area in **b**. The detailed map shows the sample locations on the RGR along the Cruzeiro do Sul Rift (CdSR) and Jean Charcot Seamount Chain (JCSC), magnetic isochrons C34n and C30n (solid white lines)[70] and estimated ocean crust isochrons (dashed white line[71];). The approximate areal extents of the groups are indicated by colored dashed lines. Bathymetric maps from: http://www.geomapapp.org[72].

hotspot track connected to a flood basalt event. Recent geochemical, geochronologic and seismic studies provide strong evidence that it was formed by a deep-rooted mantle plume (e.g.[5,14,17–19]). The plume head stage is represented by the Paraná and Etendeka LIPs on South America and Africa, respectively, formed before the continents

separated in the Early Cretaceous. After the opening of the South Atlantic, the Tristan-Gough plume formed an age-progressive submarine volcanic track (plume tail stage). It comprises the Walvis Ridge on the African Plate and its conjugate the Rio Grande Rise on the South American Plate, which formed by plume-ridge interaction ([20,21]),

followed by the younger Guyot Province, a broad volcanic province consisting of intraplate seamounts, ridges and guyots (Fig. 2a). The change from plume-ride to intraplate volcanism was linked to the initiation of geochemical zonation over the last 70 Ma[14,15]. The spatial geochemical zonation begins near the southwest end of the Walvis Ridge (DSDP sites 527, 528 & 525 transect) and continues through the Guyot Province to the active volcanic island groups of Tristan da Cunha and Gough (Fig. 2a;[14,15,22]). The geochemical distinction of the two enriched mantle one (EMI) flavors of the two sub-tracks is based on Pb isotope systematics: the two sub-tracks form a common compositional array on the thorogenic, but separate arrays on the uranogenic Pb isotope diagrams, with the Gough-type having higher $^{207}Pb/^{204}Pb$ ratios at a given $^{206}Pb/^{204}Pb$ compared to the Tristan-type[14,15]. The Gough-type geochemical signature has been traced along the entire Walvis Ridge to the Etendeka flood basalt province[14], whereas only rare Tristan-type lavas are present northeast of the DSDP transect (Fig. 2a). The geochemical zonation of the hotspot track on the African Plate over ~1600 km in separate sub-tracks has previously been explained by sampling the southwest margin of the African Large Low-Shear-Velocity Province (LLSVP; Gough composition) and the ambient mantle outside the LLSVP (Tristan composition since ~70 Ma) at the base of the lower mantle[14,15]. It remains unclear, however, whether the geochemical zonation extends further into the past and has simply not been detected thus far, because the Tristan-type material was primarily erupted on the South American instead of the African Plate. The few radiogenic isotope data from the RGR available thus far are ambiguous and could either be interpreted to reflect Gough-type[14] or Tristan-type[22] composition, resulting in completely different implications for the Tristan-Gough plume evolution.

Although, the Tristan-Gough track has been the subject of numerous studies over the past decade (e.g.[14,15,17–19,21–28]), the Rio Grande Rise (RGR) on the South American Plate has hardly been studied, with comprehensive geochronological and geochemical information only available for two sites thus far. In general, the RGR can be sub-divided based on its morphology into the Western (W) RGR and Eastern (E) RGR (Fig. 2b). The dissection of the sub-plateaus by the Cruzeiro do Sul Rift (CdSR) suggests a multi-stage history with interplay between magmatism and tectonism[29,30]. Ar–Ar ages of tholeiitic lavas from the WRGR at DSDP Site 516 (86-80 Ma;[19,21]) most likely represent the younger history of plateau formation. Alkaline lavas, dredged from Site RC16 near DSDP Site 516 (Fig. 2b), have a similar Sr-Nd-Pb isotopic composition, and are believed to represent a late alkaline plateau stage[14]. A seamount in the nearby CdSR (dredge site RC11; Fig. 2b), dated at 46 Ma, has an alkaline composition and normal-mid-ocean-ridge basalt (N-MORB)-type isotopic signature[19,27]. The alkaline N-MORB-like volcanism is probably related to the rift formation and several studies assumed a co-genetic origin with the undated Jean Charcot Seamount Chain (JCSC[30–32]). The JCSC forms a scattered NW-SE trending seamount chain from the Sao Paulo Plateau (off-shore of the Brazilian coast) to the northwest end of the CdSR on the WRGR (Fig. 2). The recovered rocks from the ERGR have a tholeiitic composition, similar to the DSDP Site 516 samples[32] (Fig. S1), and preliminary Ar-Ar results display an eastward decrease in age along the CdSR[33]. In summary, the trace element systematics of the tholeiitic lavas from the RGR[32], the surrounding magnetic anomalies (Fig. 2a), the available Ar–Ar ages[19,21,33] and plate motion models (e.g.[21,34]) are consistent with a co-genetic origin of the RGR with the Walvis Ridge through plume-ridge interaction. Although the EMI-type composition of samples from the RGR and Walvis Ridge are generally interpreted to reflect the composition of the Tristan-Gough plume[14,20,22], the origin of the RGR has remained controversial with some studies suggesting that it may be underlain by continental fragments stranded during the opening of the South Atlantic and partly covered by later plume-related volcanism as suggested by dredged continental rocks from the Western RGR[31,35].

In this work, we present Sr-Nd-Pb-Hf isotope data from five dredge sites on the WRGR and six dredge sites on the ERGR collected along the CdSR during research cruise MSM-82 with RV MARIA S. MERIAN (Fig. 2b). The study is supplemented by five samples from Deep Sea Drilling Project (DSDP) Site 516 on the WRGR and five sites on the JCSC (Fig. 2b). We show that: (1) there is no evidence for a continental fragment beneath the RGR, (2) the isotope geochemistry of the lavas supports the zoned plume model, (3) the geochemical zonation can be extended further into the past until ~100 Ma, and (4) there is evidence for compositionally distinct upper mantle plumelets since ~100 Ma. Finally, we evaluate the origin of the Tristan- and Gough-type EMI components.

## Results and discussion
### Geochemical results
As a follow-up to the major and trace element study of Hoyer, et al.[32], we subdivide the sampled structures into the following groups: (1) WRGR tholeiitic rocks from DSDP Site 516, (2) WRGR alkaline samples along the CdSR, (3) ERGR tholeiitic basalts, and (4) JCSC alkaline samples (Fig. 2b). In contrast to the tholeiitic basalts from DSDP Site 516 and the ERGR with Nb/Y < 0.8, the WRGR and JCSC alkaline lavas (alkali basalts to mugearitic rocks) have Nb/Y > 1.5. On the Nb/Yb vs $TiO_2/Yb$ diagram, the RGR and JCSC samples extend from alkalic ocean island basalt (OIB) to Enriched (E-)MORB composition, indicative of plume-ridge interaction (Fig. S1).

All our samples have isotope compositions characteristic of EMI-type lavas with radiogenic $^{87}Sr/^{86}Sr_{60Ma}$ (0.7038-0.7067), unradiogenic $^{143}Nd/^{144}Nd_{60Ma}$ (0.51215–0.51269) and $^{176}Hf/^{177}Hf_{60Ma}$ (0.28242–0.28297) and radiogenic $^{207}Pb/^{204}Pb_{60Ma}$ (15.41–15.54) and $^{208}Pb/^{204}Pb_{60Ma}$ (37.64–38.61) at a given $^{206}Pb/^{204}Pb_{60Ma}$ ratio (17.23–18.18). The new isotope data from DSDP Site 516 are within error of the previously reported ratios for samples from this site[14] (Fig. 3; Supplementary Data). The alkaline WRGR lavas lie partly within the compositional array of the previously reported EM I-like samples (site RC16[14]), but trend to a more pronounced EMI-type signature with higher Sr and lower Nd-Hf isotope ratios (Fig. 3). The tholeiitic ERGR samples overlap the WRGR DSDP Site 516 samples, but display a significantly larger compositional range. The WRGR samples largely overlap the Pb isotopic composition of the ERGR, but generally have more enriched Sr-Nd-Hf-isotopic compositions. The JCSC samples overlap the WRGR samples on the uranogenic Pb, Sr vs. Nd and Nd vs. Hf isotope diagrams, but have slightly lower $^{208}Pb/^{204}Pb_{60Ma}$ at a given $^{206}Pb/^{204}Pb_{60Ma}$ on the thorogenic Pb isotope diagram.

The data form roughly linear correlations on $^{87}Sr/^{86}Sr_{60Ma}$ vs. $^{143}Nd/^{144}Nd_{60Ma}$, $^{143}Nd/^{144}Nd_{60Ma}$ vs. $^{176}Hf/^{177}Hf_{60Ma}$ and Pb isotope diagrams, largely overlapping the Tristan- sub-track field (Fig. 3). On Sr-Nd-Hf isotope ratio diagrams, the alkaline WRGR and JCSC samples trend to a more pronounced EMI-type signature relative to the tholeiitic rocks from the RGR and Walvis Ridge. On the uranogenic Pb isotope ratio diagram, all samples plot within the Tristan compositional array, on the boundary between the Tristan and Gough fields or along an extension of the Tristan field to lower $^{206}Pb/^{204}Pb_{60Ma}$ ratios (Fig. 3).

### Does continental crust underlie the Rio Grande Rise?
Before we compare the radiogenic isotope fingerprint of the analyzed rocks to other EMI-type localities, we need to evaluate the potential role of continental crust within the RGR[31,35], which could contaminate mantle melts passing through it. In general, the EMI-type signature is consistent with a lower continental crustal geochemical affinity (Fig. 3), which could result from crustal assimilation. Alternatively, the EMI-type signature could be a deep source feature of the magmas and the reported continental rocks recovered from the flat portion of the WRGR[35] are likely to be glacial drop-stones[32].

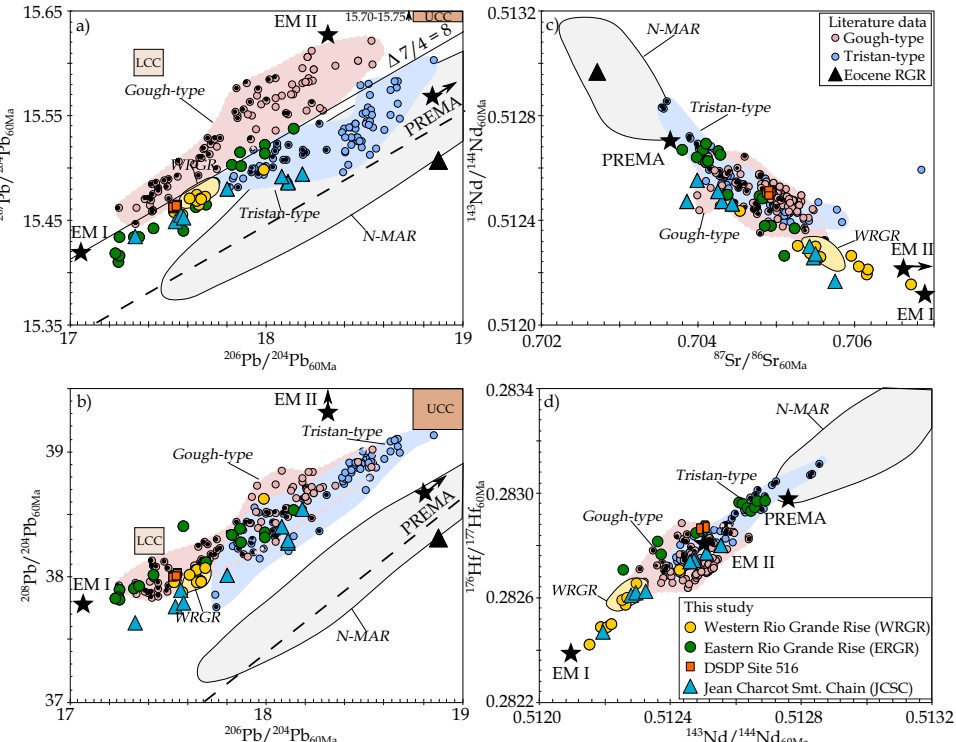

**Fig. 3 | Initial isotope ratios projected to 60 Ma show that RGR and JCSC have Tristan-type composition.** The Tristan and Gough EMI flavors can be separated on **a** the uranogenic diagram, whereas the two EMI-flavors have similar compositional arrays on **b** the thorogenic diagram, and the Gough-type have a more restricted compositional range compared to the Tristan type on **c** Sr vs Nd and **d** Nd vs Hf isotope ratios diagrams. Black dots mark tholeiitic EMI-type samples with Nb/Y < 0.8, which are restricted to the oldest portion of the Tristan-Gough track (i.e., Walvis Ridge and RGR) and have overall lower Pb-isotope ratios compared to those in the younger alkaline rocks. Dashed black line in **a, b** is the Northern Hemisphere Reference Line (NHRL[36]). The black line in **a** is Δ7/4 = 8, which almost completely

separates the Tristan- and Gough-type fields. Black stars indicate the approximate composition of the enriched mantle one, two, and prevalent mantle (PREMA) end member after Zindler and Hart[49]. Literature data from the WRGR (Late Cretaceous DSDP Site 516; dredge location RC16 = WRGR; RC11 dredge site = Eocene RGR) and the Tristan-Gough seamount chain are reported in Supplementary Data. For comparison, North Atlantic Mid-Ocean-Ridge basalts (N-MAR; compilation of ref. 62) are shown. Brownish boxes in **a, b** indicate the lower (LCC) and upper (UCC) Namibian crust after Thompson, et al.[73]. Estimated source ratios for projecting initial isotope ratios to 60 Ma to compare different aged rocks from Homrighausen et al.[27].

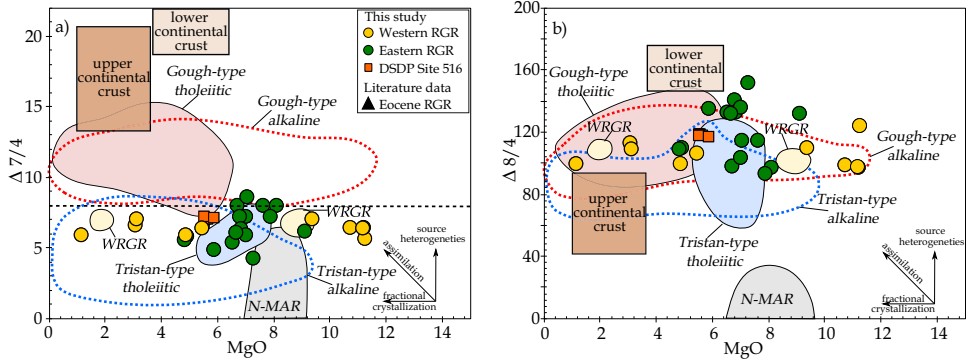

**Fig. 4 | MgO vs. Δ7/4 and Δ8/4 illustrating that the RGR lavas show no clear evidence for shallow crustal assimilation.** The WRGR data show a large spread in MgO but restricted variation in **a** vs Δ7/4 and **b** vs Δ8/4[36], forming flat arrays. Literature data are reported in Fig. 3 and Supplementary Data.

If continental material is embedded within the RGR, in particular the more evolved samples (formed by differentiation in shallow magma reservoirs) could provide geochemical evidence for assimilation of continental lithologies. The Pb isotopes in mantle melts are especially sensitive to continental crustal assimilation during fractional crystallization (AFC), due to high Pb concentrations and Δ7/4 and Δ8/4 values (deviation from the Northern Hemisphere Reference Line (NHRL) on the uranogenic and thorogenic Pb isotope diagrams[36]; Fig. 3) in continental material compared to mantle-derived magmas.

Our samples form horizontal arrays on plots of MgO (indicator of differentiation) vs. Δ7/4 and Δ8/4, indicating no AFC over a wide range of differentiation (Fig. 4). Since some of the samples are very primitive, we conclude that the enriched isotopic composition reflects a mantle source characteristic, consistent with interpretation of the trace element data[32].

An observed negative Bouguer anomaly at the RGR[31] also does not require continental crust, but could reflect overthickened oceanic crust[37,38]. Furthermore, it seems highly unlikely that countless smaller

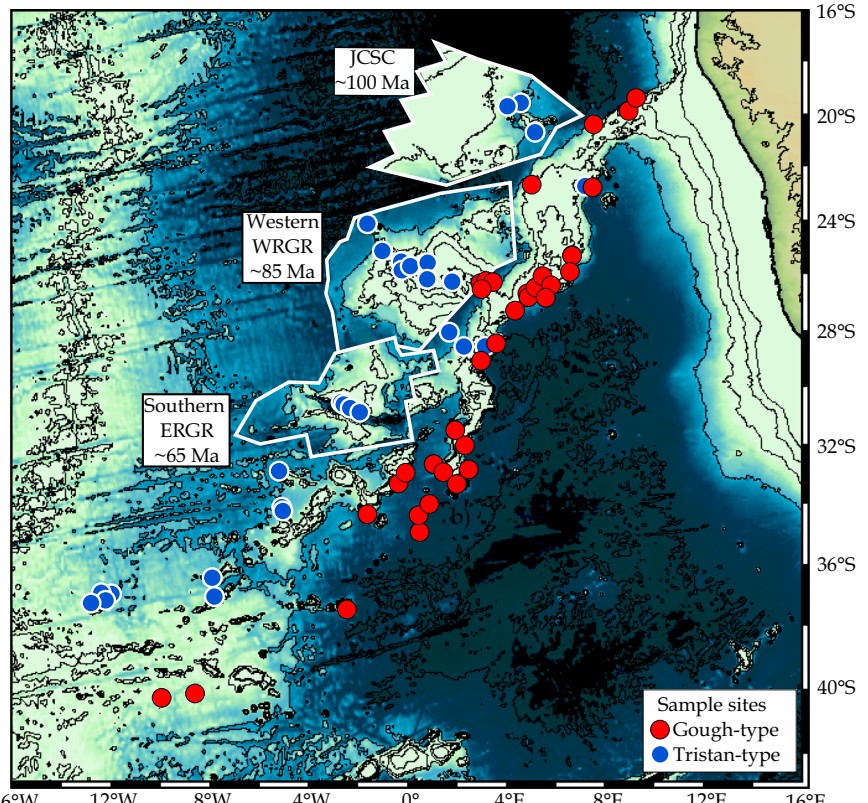

**Fig. 5 | Simplified juxtaposition of the RGR/JCSC and Walvis Ridge illustrating the large-scale geochemical zonation of the Tristan-Gough hotspot track.** The approximate locations of the RGR and JCSC when they formed according to the plate motion model from Müller, et al.[34]. A more detailed reconstruction of the temporal evolution of Tristan-Gough plume system is given in Fig. 6. Bathymetric map from: http://www.geomapapp.org[72].

continental fragments were incorporated within the oceanic crust along the entire Tristan-Gough hotspot track covering thousands of kilometers on the African and South American Plates to account for the EMI-type signature of lavas formed over ~115 Ma (Fig. 2). Our data thus support the model that the RGR and Walvis Ridge formed together as a paired "aseismic ridge", as a result of interaction of an EMI-flavored Tristan-Gough mantle plume with the young mid-Atlantic Ridge (MAR; e.g.[14,20–22,25,32]).

### The EMI-type flavor of the RGR and JCSC
The EMI-type flavors of the Tristan-Gough seamount chain form distinct fields on the uranogenic Pb isotope diagram (Fig. 3a), whereas the Sr-Nd-Hf-$^{206}Pb/^{204}Pb$-$^{208}Pb/^{204}Pb$ isotope ratios largely overlap[14,15,22] (Fig. 3b-d). Our samples have overall lower $^{207}Pb/^{204}Pb_{60Ma}$ ratios at a given $^{206}Pb/^{204}Pb_{60Ma}$ compared to Gough-type EMI and lie within the previously reported Tristan-type compositional array or extend to less radiogenic Pb isotope ratios (Fig. 3a). It is worth noting that both the Tristan- and Gough-type lavas evolve to more radiogenic Pb isotopic compositions with decreasing age and that the lowest $^{206}Pb/^{204}Pb_{60Ma}$ ratios ( < 17.6) were recovered from the oldest ( > 70 Ma) Gough-type samples[14,20], whereas previously reported Tristan-type samples (with more radiogenic Pb isotope ratios) are younger. It remains unclear if the temporal evolution to higher $^{206}Pb/^{204}Pb_{60Ma}$ ratios represents a change from plume-ridge interaction (Walvis Ridge) to intraplate volcanism (Guyot Province), or a changing plume source composition with time[20]. The RGR samples, which represent the oldest Tristan-type lavas reported so far, appear to extend this trend to even more unradiogenic $^{206}Pb/^{204}Pb_{60Ma}$ ratios. Therefore, we conclude that the RGR and JCSC have a Tristan-type composition and that both Gough- and Tristan-type lavas evolved systematically to more PREMA/FOZO-like compositions through time[20].

### Spatial geochemical zonation of the Tristan-Gough hotspot track
The distinct Tristan and Gough EMI-type compositional fields on the African Plate display a clear geographic zonation[14,15] (Fig. 2a). Gough-type lavas have been found on the Walvis Ridge and the Gough sub-track, whereas Tristan sub-track samples have exclusively Tristan-type composition. The Tristan-type composition of the RGR lavas traces the Tristan-type composition back to at least 86 Ma[19,21]. The JCSC formation is usually associated with the Eocene volcanism on the WRGR and Cabo Frio at the Brazilian coast[32], located at the NW extension of the JCSC. Multi-beam bathymetric mapping revealed that several sampled volcanic edifices along the JCSC have guyot-like structures[39] and thus can be used to roughly estimate the age since erosion to sea level. Using the depth of the erosional platforms ( -1500) the estimated age of the underlying oceanic crust (110–100 Ma; Fig. 2b) and assuming normal subsidence of the oceanic crust, we estimate that the oldest Juan Charcot Seamounts formed -10–20 Ma after their underlying ocean crust indicating a Late Cretaceous formation (Fig. S3). Our plate reconstruction (Figs. 5 and 6) illustrates that both the RGR and JCSC were once located adjacent to the Walvis Ridge and that the spatial geochemical zonation can nearly be traced along the entire combined submarine hotspot track (Fig. 5).

Geochemical zonation is a widespread phenomenon found in numerous plume systems, including the Hawaiian and Galapagos hotspot tracks (e.g.[13,16,40]). The longevity of the geochemical zonation of the Tristan-Gough hotspot track, however, seems unique. At the Hawaiian Islands, for example, the Loa and Kea compositional types display two sub-parallel spatial trends over the last ≤6 Ma[11,41]. The older part of the Hawaiian hotspot track, including the Emperor Seamounts and Hawaiian Ridge (-85-6 Ma), is dominated by the "depleted" Kea composition without a clear spatial zonation and only

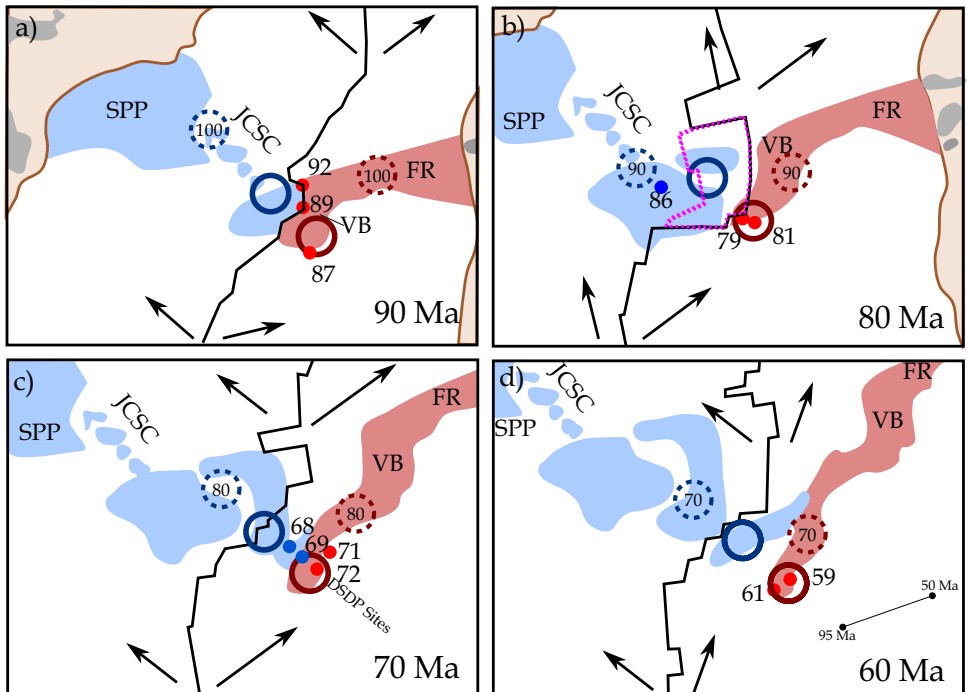

**Fig. 6 | Schematic reconstruction of the temporal evolution of Tristan-Gough plume system using GPlates.** Our model is shown at **a** 90 Ma, **b** 80 Ma, **c** 70 Ma, and **d** 60 Ma. The arrows in the model[34] indicate the direction of plate motion. The magmatic structures are color-coded to the reported and assumed (i.e., SPP and large parts of the RGR) pre-dominant Tristan-type (blue) and Gough-type (red) composition. Large open blue (Tristan) and red (Gough) circles indicate the proposed position of the Tristan and Gough plumelets (see also Figs. 2 and 7) and dashed circles the relative position of the respective plumelets 10 Ma earlier. Small dots display Gough-type (red) and Tristan-type (blue) volcanism at a given time[17,19,21,22]. The dashed field in **b** indicates the approximate area of the proposed microplate[24,26]. The MAR shape is from Müller et al.[34], except at 80 Ma, which is from Sager et al.[24]. In **d** the plume motion is indicated (black line labelled 95 and 50 Ma), which is required for the Müller et al. (2019) model to account for the older portion of the Tristan-Gough plume system. FR Frio Ridge, RGR Rio Grande Rise, JCSC Jean Charcot Seamount Chain, SPP Sao Paulo Plateau, and VB Valdivia Bank.

sporadic occurrence of the more enriched Loa composition between ~47-6 Ma[42]. Spatial zonation of the Galapagos hotspot track, on the other hand, can be traced back to ~20 Ma[13] and possibly was present ~60–90 Ma ago based on data from accreted Galapagos terranes in Central America[40].

In general, hotspot track geochemical zonation has been explained by: (1) an axially-asymmetric, laterally-zoned plume conduit that preserves heterogeneities sampled at the root of the plume through laminar flow (e.g.[12–16,40,42,43]), or (2) an unzoned (randomly heterogeneous) plume consisting of enriched fertile lithologies (e.g., pyroxenite/eclogite) within a relatively depleted peridotitic matrix with lateral variations in melting conditions, such as temperature and pressure (e.g.[9,10]). In the case of the Tristan-Gough hotspot track on the African Plate, variable melting conditions cannot adequately explain the geochemical zonation (see Hoernle, et al.[14]). The same is true for the RGR, which shows that the Tristan-type signature is recorded by tholeiitic and alkaline rocks formed under very different melting conditions[32]. The long-lived ( ≥100 Ma) and large-scale spatial geochemical zonation along the Tristan-Gough system can thus be best explained by geographically-separated and geochemically distinct source compositions.

**Magmatic evolution of the RGR and JCSC: Evidence for distinct upper mantle plumelets?**
While the temporal evolution of the studied structures remains in part speculative, the spatial arrangement of the geochemical signature could have important implications for the dynamics of the Tristan-Gough mantle plume system (Fig. 5). Although the exact emplacement time of the alkaline intraplate volcanism at the WRGR and JCSC remains unclear, the presence of the Tristan-type material along the entire JCSC seems enigmatic. How did the Tristan-type material, which has so far

only been reported from structures closely associated to the Tristan-Gough plume system, get up to 500 km NW of the Rio Grande Rise?

We now present our preferred model to explain the spatial arrangement of the geochemical signatures. Our proposed model is based on the v2 plate motion model[34] and provides an excellent reconstruction for the Tristan-Gough track until ~70 Ma. Before 70 Ma, plume motion is required to reconstruct the formation of the Walvis Ridge and Parana-Etendeka LIP, which is (to our knowledge) required for all plate motion models. To begin with, it has been proposed that the Tristan and Gough EMI-type lavas were derived from distinct Tristan and Gough upper mantle plume conduits or plumelets to explain the geochemical zonation over the last ~70 Ma and the >400 km distance between the active island groups[19,22] (Fig. 7b). A regional seismic tomographic study imaged a narrow (~200 km diameter) conduit to the SW of the Tristan island group extending to ~500 km depth, confirming the presence of a plumelet beneath an extension of the Tristan sub-track[18]. Unfortunately, no detailed seismic studies have been carried out around Gough Island, but it is assumed that a similar small plume conduit (plumelet) is feeding the Gough sub-track[19,22] (Fig. 7b). Global seismic tomography only shows a large, dome-like upwelling extending from the deep mantle to about 1000 km depth beneath the Tristan-Gough region[5], suggesting a bifurcation into distinct upper mantle plumelets above 1000 km, which cannot be resolved by the current global models. Such a fork-like or plume-tree structure has been recently reproduced by numerical models[7]. We assume that the recent spatial arrangement of the observed Tristan and expected Gough plumelets in the upper mantle remained relatively constant throughout the plume tail stage.

In our model, we fix the expected Gough plumelet to volcanism with a Gough-type signature at a given Ar-Ar age along the Tristan-Gough track on the African plate[17,19,22] to constrain the possible

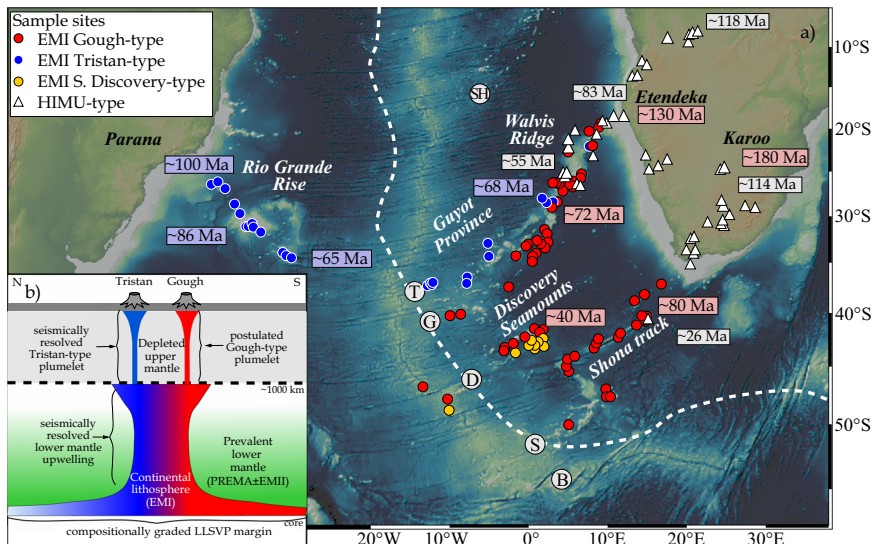

**Fig. 7 | Composition of the South Atlantic hotspot tracks and schematic cartoon of the Tristan-Gough tree-like plume structure and compositionally graded LLSVP. a** The color-coded sample sites display the geochemical fingerprint of the South Atlantic hotspot tracks. The approximate age of the hotspot lavas decreases progressively towards the ridge axis, illustrating the longevity and large volume of the geochemical types. The dashed white line indicates the margin of the African LLSVP[74] showing that the EMI-type Tristan (T), Gough (G), Discovery (D) and Shona (S) hotspots overlie the margin of the LLSVP, whereas the PREMA-/FOZO-type Bouvet (B) hotspot lies outside the LLSVP. Literature data: Tristan-Gough track is reported in Fig. 2, Discovery[45,75] and Shona[76,77]. **b** The colors indicate geochemical grading of the southwest margin of the LLSVP and predominant geochemical composition of the upper (DM) and lower mantle (PREMA ± EMII). Laminar flow (e.g[78]) preserves the geochemical grading in the lower mantle upwelling[5] to a depth of ~1000 km. The plumelets rise from the top of the compositionally graded lower mantle domal upwelling[18,19] with distinct geochemical composition. Source of bathymetric map: http://www.geomapapp.org[72].

position of the Tristan plumelet. We use the oldest reported age of a given site, which represents a minimum age of the exposed hotspot volcanism. Using these assumptions, the Gough-type plumelet resided beneath the African Plate since at least 115 Ma[15,19], whereas the Tristan-type conduit was located beneath the South American Plate until ~70 Ma (Fig. 6). Thereafter, the model predicts that the African Plate also moved over the Tristan plumelet, which is consistent with the initiation of the observed geochemical zonation on the African Plate reported for the Leg 74 DSDP Site transect (Fig. 2 & 6). Below we describe our model in more detail.

At ~115-90 Ma, the Gough plumelet lay beneath the young MAR and formed the Frio Ridge (FR; the oldest part of the Walvis Ridge; Fig. 6a). The projected Tristan plumelet was located beneath the South American Plate and could have formed the Sao Paulo Plateau and subsequently the JCSC (Fig. 6a), which is consistent with the relative NW motion of South America, the orientation of these structures, our age estimates, and the Tristan-type geochemical signature of the JCSC. We, therefore, favor formation ages of ~100-90 Ma for the JCSC. Even if the JCSC is younger than our estimated Cretaceous age (Fig. S3), our model could explain the occurrence of Tristan-type material 500 km NE of the RGR, if a low volume Tristan plumelet was located beneath the oceanic lithosphere underlying the JCSC. Plume-ridge interaction or at least flow from the Tristan plume to the MAR could have formed the western portion of WRGR (Fig. 6a).

Between ~90-80 Ma (Fig. 6a, b), the Gough plumelet was located beneath and formed the Valdivia Bank (Fig. 6a, b), whereas the Tristan plumelet was located beneath the WRGR and then beneath the ERGR (Fig. 6a, b). The formation of the WRGR due to plume-ridge interaction of the Tristan plumelet is supported by the Tristan-type composition of the WRGR. Recently it has been proposed that a microplate, indicated by the dashed field in Fig. 6b-d, was located between the Valdivia Bank and WRGR around this time[24,26]. The exact dimensions and shape of the microplate are unclear, but could explain the formation of both the northern ERGR and eastern portion of WRGR by plume-ridge interaction or flow from the Tristan plumelet to the western microplate boundary, assuming an outline as shown in Fig. 6b. The existence

of this microplate could also explain the clear spatial separation of the Tristan- and Gough-type material by interaction of the respective plumelets with the western and eastern sides of the microplate, respectively, although more sampling is required to determine the composition of the northern parts of the RGR.

From ~80 to 70 Ma, the Gough plumelet formed the Walvis Ridge extending to DSDP Site 525 (Fig. 6c, d). Due to the ongoing westward migration of the MAR, the supply of plume material to the ridge eventually ceased and the Gough plumelet became an intraplate hotspot. Likewise, the WRGR migrated away from the Tristan plumelet, which now supplied material to the MAR to form the central and southern portion of the ERGR (Fig. 6c, d). At ~70 Ma, several eastward directed ridge-jumps finally transferred the entire RGR to the South American plate[21]. After 70 Ma, the changing MAR geometry and its westward drift resulted in the African Plate moving over the Tristan plumelet and formation of DSDP Sites 527 & 528 (Fig. 6c). At ~65 Ma, the compositionally distinct Tristan and Gough sub-tracks formed on the African Plate above the two plumelets (Fig. 6d).

Although the exact location of the MAR, the dimension of the microplate and assumed plume-ridge interaction are somewhat speculative, this model can explain the overall spatial geochemical distribution of Tristan- and Gough-type volcanism on both the African and South American plates. The different volumes/areal extents of the respective structures are also consistent with the model. The small JCSC volcanic edifices were most likely emplaced in an intraplate setting, whereas the voluminous RGR plateau and Valdivia bank formed when the respective plumelets interacted with the spreading ridges (Fig. 6).

Importantly, the Tristan composition is much more widespread than previously believed and may extend as far back as the breakup of Africa and South America. As noted above, the Sao Paulo plateau could represent the Tristan-type counterpart of the Gough-type Frio Ridge (Fig. 6), extending the geochemical zonation to the opening of the South Atlantic. In this case, the appearance of the Tristan-type composition would coincide with the change from plume head to plume tail stage and indicate that the bifurcated mantle upwelling with two

distinct upper mantle plumelets remains relatively stable throughout the entire plume tail stage. The structural change from plume head to tail stage could be caused by a viscosity change between the lower and upper mantle at ~1000 km depth[6,7]. Whereas the massive and buoyant starting plume head was able to punch through this rheological barrier, the subsequent upwelling plume material largely stalled at ~1000 km depth with only smaller plumelets managing to find pathways through this boundary[7]. This scenario, however, is highly speculative and needs to be confirmed by rheological modelling, as well as the effect of mantle wind, which could lead to motion and/or dispersion of the relatively low-volume plumelets[7]; however, local seismic tomography clearly imaged a rising plumelet just southwest of the Tristan island group[18]. Additionally, recent buoyancy flux estimates (estimated plume flux divided by two for the proposed plumelets), however, are comparable to the estimates from long-lived plume systems, such as the St. Helena, Meteor/Shona or Louisville hotspot tracks[44]. Finally, we believe that our model provides a new perspective of the complex temporal and geochemical evolution of the oldest submarine portion of the Tristan-Gough hotspot track, but further sampling is required, especially at the transition of the plume head to plume tail stage (e.g. by drilling into Sao Paulo Plateau), to fully understand the plume dynamics and geochemical variations.

### New insights into the origin of Tristan-type EMI and of the geochemical zonation of South Atlantic plumes and their sources

As discussed above, the Tristan-type EMI composition persisted most likely throughout the plume tail stage and is the dominant component at the RGR. Thus, the Tristan-type volcanism is similar in both temporal and areal extent to the Gough-type volcanism in the hotspot track on the African Plate (Fig. 2 & 5). Because the Gough-type is associated with elevated $^3He/^4He$ isotope signatures (see summary in Homrighausen, et al.[22]), it is believed to be derived from the margin of the Large Low-Shear Velocity Province (LLSVP) at the base of the lower mantle[14,15,22,45,46]. What, however, is the origin of the Tristan EMI-type composition?

Three distinct EMI-type flavors (Tristan, Gough and southern Discovery) have been identified in the zoned Tristan-Gough, zoned Discovery (northern Discovery has a Gough-type and southern Discovery an even more enriched composition) and Shona (Gough composition) hotspots (Fig. 7a). All samples within one of the three compositional groups form linear arrays on Sr-Nd-Hf and thorogenic Pb isotopic correlation diagrams (e.g., Fig. 3), as do EMI lavas globally[47]. The arrays can be largely explained by mixing of two mantle endmembers: EMI and prevalent mantle (PREMA) or the focal zone for oceanic lavas (FOZO) with similar compositions[48,49], perhaps with a minor additional contribution of EMII (enriched mantle two[49]; Fig. 3)[22]. The origin of the EMI component is generally related to delaminated ancient metasomatized subcontinental lithospheric mantle (SCLM) and/or lower continental crust (e.g[20,25,47,50]), whereas the PREMA/FOZO components are believed to represent ambient lower mantle[45,49], possibly containing some EMII representing either subducted marine sediments or upper continental crust[49] (Fig. 7b).

In contrast to the other radiogenic isotope systems, the relationship between the two uranogenic Pb isotope systems is complex in EMI magma sources, because the half-life for the decay of $^{238}U$ to $^{206}Pb$ ($t^{1/2}$ = 4.47 Gy) and $^{235}U$ to $^{207}Pb$ ($t^{1/2}$ = 0.7 Gy) are significantly different. Thus, most $^{207}Pb$ was formed during early Earth history (Archean or older), whereas presently most remaining U is $^{238}U$, so that primarily $^{206}Pb$ is formed presently. U/Pb fractionation events happening at different times in Earth history will therefore result in different uranogenic Pb growth curves, generating different source compositions. As a result, ancient EMI sources that formed by similar processes at different ages or with different U/Pb ratios can have broadly overlapping Sr-Nd-Hf-$^{208}Pb$ isotope ratios, but have distinct compositions on the uranogenic Pb isotope diagram[47], explaining the difference in

$^{207}Pb/^{204}Pb$ ratios at a given $^{206}Pb/^{204}Pb$ ratio in the different South Atlantic EMI flavors (Tristan lowest, Gough intermediate and South Discovery highest).

Both Tristan-Gough and Discovery hotspots have bilaterally zoned hotspot tracks (Fig. 7a), suggesting that the plumes feeding them are also bilaterally zoned. The most popular explanation for the bilateral zonation is the sampling of different portions of the lowermost mantle with one being from inside the LLSVP (e.g. Gough; Loa at Hawaii) and the other from the ambient mantle surrounding the LLSVP (e.g. Tristan; Kea at Hawaii)[14–16,42]. Since LLSVPs have distinct physical properties compared to the ambient lower mantle (temperature and/or composition)[51] and recent numerical models indicate that bilateral plume zonation requires similar physical properties for both sources[52], both compositional types should either be derived from the ambient mantle or from the LLSVP. Chemical heterogeneity within one of these reservoirs explains the NNW-SSE oriented geochemical zonation better than sampling the two sides (inside vs. outside) of the N-S oriented LLSVP boundary beneath these hotspots, as previously proposed (Fig. 7a). Interestingly, all 13 oceanic hotspots with EM-type composition are geographically associated with LLSVPs, whereas hot spots located far from LLSVPs exhibit exclusively non-EM compositions[53]. The Bouvet hotspot, south of the Shona hotspot (located outside the projected LLSVP margin)[53], has a PREMA/FOZO-type composition with high $^3He/^4He$ isotope ratios, and thus most likely represents the ambient lower mantle beneath the South Atlantic[45]. In light of these observations, we propose that the different EMI flavors represent delaminated SCLM ( ± lower continental crust) with different ages of U/Pb fractionation, which were stored within the LLSVP until they were sampled by mantle plumes.

There is a broad trend in Atlantic MORB and OIB compositions towards higher Δ7/4 going from north to south[47]. We propose that this large-scale geochemical gradient could ultimately reflect latitudinal compositional grading of the African LLSVP, resulting from accumulation of delaminated SCLM ( ± lower continental crust) with different enrichment ages and/or at different times during Earth history. The SCLM could have had progressively different metasomatic ages and/or U/Pb ratios with latitude, e.g., reflecting a southward migration of subduction initiation in early Earth history[47]. In conclusion, our study suggests that large-scale geochemical grading of the LLSVP, resulting from hemispheric-scale events in early Earth history, can be preserved in complex upwelling plume structures, leading to spatial geochemical hotspot track zonation at the earth's surface (Fig. 7).

## Methods
### Sample preparation
The sample locations and descriptions are reported in detail by Geissler, et al.[39] and Hoyer, et al.[32]. The samples recovered during the cruise MSM-82 with RV Maria S. Merian were cut on board into blocks with a rock saw to remove the most altered outer portions of the rocks. The fresh-looking interiors of the rocks were cleaned in an ultrasonic bath with deionized water and afterwards dried at 60 °C for 12 h. The blocks were then crushed with a hydraulic press, sieved, washed in deionized water in an ultrasonic bath and dried in an oven at 60° for 12 h again. To minimize secondary materials and altered rock portions, the rock chips were carefully handpicked under a binocular microscope. The major and trace element analysis of the Jean Charcot Seamount and Rio Grande Rise samples are reported by Hoyer, et al.[32].

Sr–Nd–Pb–Hf radiogenic isotope analyses were conducted at GEOMAR Helmholtz Centre for Ocean Research Kiel, Germany by thermal ionization mass spectrometry (TIMS, Sr–Nd–Pb) and MC-ICPMS (Hf). Chemical preparations were carried out in Class 10,000 laboratories equipped with MK® Class 100 vertical laminar flow work stations and PicoTrace® Class 10 horizontal laminar flow fume hoods. Reagents used are ultra-pure HF and HBr from Seastar® purchased via

Roth® and in-house double distilled HCl and HNO₃ using a PicoTrace© cascading distill and finally ultra-pure 18.2 MΩ H₂O is made by an Elga LabWater® Purelab device fed by deionized H₂O.

In order to minimize the effects of alteration and contamination introduced during handpicking of the rock chips for Nd-Pb-Hf isotope analysis, ~100–200 mg of rock chips were acid-washed with 2 N HCl at 70 °C for 2 h and subsequently triple-rinsed in 18.2 MΩ H₂O. For Sr isotope analysis, 100–200 mg of rock powder was leached in 6 N HCl at 130 °C for 3 h and then triple rinsed in 18.2 MΩ H₂O to remove seawater Sr (with $^{87}Sr/^{86}Sr = 0.7060–0.7090$ and Sr concentration of ~8 ppm). To ensure liquid-solid separation, sample vials were centrifuged at 3095 g for ten minutes in an Eppendorf® 5810 centrifuge equipped with customized holders for 15 ml Savillex® Teflon beakers. 1.5 ml of conc. HNO₃ and 4 ml of conc. HF were added to the leached sample residue which was then dissolved for 48 h in closed 15 ml Savillex beakers at 130 °C. The sample beakers were then placed in an ultrasonic bath for 30 minutes to ensure complete sample digestion and then dried at 130 °C. Thereafter 5 ml of 6 N HCl were added to the sample residue for chloride complexation in closed beakers overnight at 130 °C on the hotplate. Another 30 min in an ultrasonic bath and visual inspection for complete dissolution were carried out before samples were dried again. Sequential Pb-Sr-Nd separation basically followed procedures of Hoernle et al.[54] into which Hf separation was integrated. They comprise a two-pass Pb separation and clean-up using 100 µl Teflon micro-columns filled with Bio-Rad AG 1 × 8 (100–200 mesh) anion resin equilibrated with 1 M HBr for the highest Pb retention and Pb release with 1 ml of 6 M HCl. The rare-earth elements (REEs) were obtained over 6 ml 6 M HCl at the final washout during Hf–Rb–Sr separation on quartz glass columns filled with Bio-Rad AG50W-X8 (100–200 mesh) cation resin. The REEs were then loaded in 0.22 M HCl onto 4 ml quartz glass columns filled with EICHROM® Ln-Spec resin (100–150 µm) to obtain the Nd fraction. Sr of the strongly leached powders was separated by a single pass on one-time BioRad micro columns filled with 0.2 ml of TRISKEM® SR-Resin (50–100 µm) that was equilibrated (1.5 ml) and rinsed (2.5 ml) with 6 M HNO₃ and Sr being washed out in 1 ml of 0.05 HNO₃. Hf was separated on the cation exchange column over the first 2 ml using a 1.5 M HCl media followed by a Hf clean-up in 0.2 ml of TODGA resin (50-100 µm; DN-B10-S TRISKEM®) after Connelly, et al.[55] 5% of the final Hf washout was measured on an Agilent 7400 ICPMS for Hf concentrations and total chemistry yields for all processed samples and the BCR-2 standards lie between 40% and 70%, which allow multiple Hf isotopic analysis of 50 to 100 ppb solutions.

## Sr–Nd–Pb–Hf isotope ratio analysis

Sr and Nd isotope ratios were determined on a Thermo Scientific® TRITON *Plus* TIMS operating in multi-dynamic collection mode and normalized within run to $^{86}Sr/^{88}Sr = 0.1194$ and $^{146}Nd/^{144}Nd = 0.7219$, respectively. Measurement errors are reported as a 2σ standard error (2SE). Single element reference materials NBS987 (Sr) and La Jolla (Nd) were measured 4 to 5 times along with the samples on each turret and the average $^{87}Sr/^{86}Sr$ and $^{143}Nd/^{144}Nd$ of the standard subtracted from the preferred reference values to obtain a delta value to be added to the sample and standard data of each turret. This procedure ensures long-term comparability of data generated at different times. Accordingly, NBS987 gave $^{87}Sr/^{86}Sr = 0.710250 ± 0.000006$ ($n = 24$; 2σ standard deviation (2 SD) and La Jolla gave $^{143}Nd/^{144}Nd = 0.511850 ± 0.000007$ ($n = 26$; 2 SD). Pb isotope analyses were carried out in static multi-collection mode on the TRITON *Plus* using the Pb double-spike (Pb-DS) technique of Hoernle et al.[56]. The long-term Pb-DS corrected NBS981 values are $^{206}Pb/^{204}Pb = 16.9408 ± 0.0018$, $^{207}Pb/^{204}Pb = 15.4974 ± 0.0019$, $^{208}Pb/^{204}Pb = 36.7206 ± 0.0048$, $^{207}Pb/^{206}Pb = 0.914799 ± 0.000049$ and $^{208}Pb/^{206}Pb = 2.167581 ± 0.000094$ ($n = 257$; 2 SD) since installation of the instrument in 2014.

Hafnium isotopic analyses were measured on a Thermo Scientific® NEPTUNE *Plus* MC-ICPMS in static multi-collection using 100 ppb Hf sample solutions and following the instrument set-up outlined in Dausmann et al.[57]. Our in-house Hf SPEX CertiPrep™ solution is calibrated with $^{176}Hf/^{177}Hf = 0.282170$ against $^{176}Hf/^{177}Hf = 0.282163$ for JMC-475 Blichert-Toft et al.[58] and was measured every 6th sample to apply instrumental drift correction within an analytical session. The normalized Hf SPEX CertiPrep™ gave $^{176}Hf/^{177}Hf = 0.282170 ± 0.000004$ ($n = 69$).

Total chemistry blanks were typically <30 pg Pb, <100 pg Sr, <50 pg Nd & Hf and are therefore considered negligible relative to the amounts of sample used. Replicate analysis of samples by means of a second digest are shown in the Supplementary Data. For Nd-Hf and Pb they are within the 2 SD's of the single element standards mentioned above except $^{208}Pb/^{204}Pb$ and $^{206}Pb/^{204}Pb$ of sample MSM82-12-DR-1. For $^{87}Sr/^{86}Sr$ deviations outside 2 SD of NBS987 are noted for MSM82-65-DR-2 and MSM82-12-DR-1. All offsets are ascribed to heterogeneities of the sample powder/chips with variable Rb/Sr, U/Pb and Th/Pb domains and extent of Sr addition from seawater upon alteration. Reference material BCR-2 were repeatedly dissolved, processed and measured. Its radiogenic Sr-Nd-Pb and Hf isotope ratios lie within the values of Fourny et al.[59] and Todd et al.[60].

## Initial and projected radiogenic isotope ratios

Since the recovered samples and the comparative literature data were emplaced over an age range of ~115 Ma, we follow the approach of Homrighausen, et al.[22] and project the calculated initial isotope ratios, using the measured parent and daughter element concentrations, to a common age of 60 Ma, using the proposed EMI parent/daughter source ratios of Willbold and Stracke[61]. The Mid-Atlantic Ridge (MAR) literature data (compiled by Class and Lehnert[62]) was back-projected to an age of 60 Ma with the proposed depleted mantle values from Workman and Hart[63]. The measured, initial, and projected radiogenic isotope ratios and age estimates for different parts of the plateau and seamounts are reported in the Supplementary Materials.

## The role of alteration

The recovered submarine rocks from the RGR and JCSC have undergone variable degrees of seawater alteration[32,39]. Even though the freshest parts of the lavas were selected and carefully hand-picked under a binocular microscope (0.5–1 mm size fraction), it was not always possible to completely remove all altered material due to pervasive effects of groundmass alteration. Overall, the loss-on ignition (LOI), a common indicator for the degree of alteration, is relatively low for such, presumably old, submarine rocks (LOI < 4.0 wt% for 28 of the 40 samples, and 4–6.8 wt% for the remaining samples)[32].

The Sm-Nd and Lu-Hf isotope systems are generally considered to be resistant to low-temperature alteration and correlate well ($^{143}Nd/^{144}Nd_{initial}$ vs. $^{176}Hf/^{177}Hf_{initial}$ ratios gives an $R^2$ of 0.91; Fig. S2). The Sr isotope ratios, on the other hand, can be significantly elevated by exchange with seawater Sr (with $^{87}Sr/^{86}Sr$ ratios of ~0.707–0.709;[64]). The relatively good $R^2$ values from plots of $^{87}Sr/^{86}Sr_{initial}$ vs. $^{143}Nd/^{144}Nd_{initial}$ (0.77) and $^{176}Hf/^{177}Hf_{initial}$ (0.71) ratios, suggest that the initial $^{87}Sr/^{86}Sr$ ratios representative of their primary magmatic ratios (Fig. S2). The Pb isotope ratios are essential to discriminate between the Tristan- and Gough-type components[14,15]. While Th is immobile, U is relatively mobile during low-temperature seafloor alteration and is generally taken up by submarine rocks from seawater. Both U and Pb, however, can be mobilized by hydrothermal fluids (e.g.[65]). Since the reported rocks display a relatively good correlation of Th vs Pb ($R^2 = 0.71$, excluding MSM82-72-DR-3 with extremely high Ce/Pb = 57; Fig. S2) and vs U ($R^2 = 0.81$, Fig. S2), we assume that Th-Pb-U concentrations largely reflect those of the

magma and that the initial radiogenic isotope ratios also represent primary magmatic ratios.

## Data availability

All data needed to evaluate the conclusions in the paper are present in the paper and/or the Supplementary Materials.

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

## Acknowledgements

We thank Captain Schmidt and his crew for their support during R/V Maria S. Merian cruise MSM82 and the scientific crew of MSM82 for

assistance with dredging and initial sample processing. We thank S. Hauff and K. Junge for analytical support and M. Gutjahr for the Hf isotope measurements at GEOMAR; C. Langmuir and G. Zhang for constructive reviews that helped improve this manuscript; and R. Neely for editorial handling. This research used samples provided by the Integrated Ocean Drilling Program. We thank the GEOMAR Helmholtz Centre for Ocean Research Kiel for providing funding for the isotopic analytical work. S. Homrighausen was funded by the Deutsche Forschungsgemeinschaft (DFG) Grant 439265336 while preparing this publication.

## Author contributions

K.H. and K.M.H., together with others conceived the project; W.H.G. was the chief scientist of the cruise; S.H., P.A.H., and J.G recovered the offshore samples; F.H. generated the isotope data; S.H. prepared the figures and paleogeographic reconstructions; S.H. and K.H. were the primary interpreters of the data and writers of the manuscript with contributions from F.H., P.A.H., K.M.H., W.H.G., and J.G.

## Funding

## Competing interests

The authors declare no competing interests.
