## [Peer Review File · Nature Communications]

REVIEWER COMMENTS

Reviewer #1 (Remarks to the Author):

This is a solid paper and contribution to knowledge, with important new observations that contribute to our understanding of mantle plumes. The paper reflects a new field of investigation, which one might call “isotope tectonics” which the authors have pioneered. While the observations of the paper are geochemical--isotopes on samples previously measured for major and trace elements-- the data are not used for geochemical modeling but for constraining mantle flow and surface tectonics. The paper provides new insights into the structure of mantle plumes, and the (surprising) longevity of those structures. The data are important and the data quality is good, the observations are new and important, the conclusions have far-reaching implications. This is a good paper worthy of publication in Nature Communications.

One could argue that the paper is “incremental” since as shown in Figure 5, a long-lived zonation was previously known on the African plate. However, the extension of the data to the Atlantic Plate and the extension in time to 100Ma are important enough in my view that the papers merits Nature communications.

Ordinarily I am a very critical reviewer, as people often use a small amount of data, sometimes of questionable quality, to come to far-reaching conclusions not necessarily well supported by the data. These authors are instead understated, perhaps to a fault. They have a clear question they proposed for a ship-based sampling program—what is the origin of the Rio Grande Rise and how long has the spatial zonation of the Tristan and Gough plume components persisted? And they come up with clear answers. The Rio Grande Rise is not a continental fragment but a manifestation of plume volcanism, and the plume assemblage in the S. Atlantic has produced spatially organized distinct compositions for one hundred million years. It’s amazing, even though the authors adopt a low key approach.

The authors propose a reasonable physical model for the origin of their observations. I suppose that is all that can be required of them, but in my view the paper would have greater impact if the authors considered some of the following--

To highlight the remarkable result, I would compare the longevity to other known zoned plumes, such as Hawaii and the Galapagos. I think such a comparison would be of great interest to most readers.

Can plumelets really last 100MA in a vigorously convecting upper mantle? Since they are relatively low volume, I do not understand why they would not become more dispersed by general mantle flow.

Mantle components are good for tectonic mapping such as is done here, but I they are really just a description, not an explanation. Perhaps it is discussed in some of their previous papers on this topic, but I wanted to know what was the cause of the differences between the two discrete EM1 components. I also wanted to know if there were trace element differences between the components

that would explain why there are uraniumogenic and not thorogenic differences, and what the origin of these components was in the Earth.

I do not think a reviewer one can require those extensions, as each one would involve significant additional work and perhaps additional co-authors such as geodynamicists, but maybe they are worth a paragraph of discussion?

Since much of the work in this region has been done by the authors, they are very familiar with it. For the average reader who might be looking at the data for the first time, a bit more background and reporting of previous work would be useful. The authors are to be commended for including the Hoyer et al data in their appendix. Having data on the Tristan and Gough end members would also be useful, particularly since many of the figures have fields for them.

The statements on lines 414ff that changing the number of samples from $n=33$ to $n=72$ might explain the difference in range is to me questionable. You could prove it by taking a random group of 72 samples and boot-strapping 33 at a time and seeing how often range is restricted. Sampling half the population usually does a good job of representing the whole. Maybe instead the difference is real? What then?

The paper is a solid piece of work that everyone working in this field should read, Even those peripheral to the field would I think find the results fascinating. With minor revisions I would recommend publication.

Charles Langmuir
Harvard University

Reviewer #2 (Remarks to the Author):

This manuscript reports extensive Sr-Nd-Pb-Hf isotopic data for seafloor volcanic rocks from the Rio Grand Rise and JCSC, and their old ages and Tristan-type isotopic signature lead the authors to suggest a long-lived zonation model for a mantle plume, based on the tectonic reconstruction. I read the paper

with interests and find this paper might constitute a good scientific contribution to understanding the nature of mantle plume if the geochemical zonation can be traced back to the plume head stage. I also find some problems that need to be addressed before it can be accepted for publication. I have some comments for the manuscript, and look forward to seeing their reply.

(1) Discussion on the role of alteration: I feel the Section “the role of alteration” is not necessary in Discussion. The first paragraph is on the processing procedure and LOI, it would be better to show this in the Results. Moreover, leaching with 6HCl would be sufficient to remove the effect of seawater alteration, this can be verified by a number of papers. The authors should have confidence on their data, thus, it is not necessary to show the correlations between these isotopes, as shown in Lines 247-253, so I suggest to remove this information from the manuscript. The sample description, including alteration characteristics, can be shown in Results.

(2) For the analytical methods: Leaching with HCl is crucial for Sr-Pb isotopes of seafloor samples, but they indicated only the leaching method for Sr, what about Pb? Leaching with with 6NCl at 150C for 3h seems too strong, it would generally be ~70C, I suggest the authors to present the trace element data after leaching if available. They should show not only the measured results of standards but also the reference values, including BCR-2. Presentation of these data are important to understand the data quality.

(3) The authors should carefully compare the present study with one of their previous publication Hoyer et al (2022)-CEE. The present paper try to publish the isotopic data separated from Hoyer et al (2022), but the two papers seem to have very similar topics, like continental contamination, tectonic reconstruction, age relationships, et al. To me, Fig. 6 of this ms is quite similar to Fig. 7 of Hoyer et al (2022). I understand the new isotopic data provide robust evidence for the plume model, but they also need to mention what have already been addressed, and what is new compared to Hoyer et al (2022).

(4) Section “Age constraints and implications for the RGR and JCSC lavas”: the title of this section is misleading here. This manuscript does not provide new data of ages, thus, the title should be revised. Also, from my point of view, they should present the information on formation ages in the Section “Geologic Overview”.

(5) For Section “Continental crust embedded in the Rio Grande Rise?”. I don’t quite understand the logic of this section. As shown in the last paragraph on the previously reported continental rocks in this area should be not be in situ, and thus, does not support the presence of continental fragments under the rise. This information should be at the very beginning of this section. Then they further discuss the possible continental contamination based on isotopic results, but it is not very convincing to exclude the possible origin of EM1 signature from continental contamination. While they propose that Pb-isotopes are good indicator of contamination, then they don’t believe the indication of Pb isotopes on the

contamination (Lines 337-340). They prefer to use MgO and Th/La to preclude the continental contamination. However, MgO is highly dependent on the magma evolution and fractionation processes. (6) The authors need to consider a more convincing way to exclude the contamination, for example, they can precisely compare their data with the corresponding crust component on the south America on their figure 3.

(7) Lines 441-452: It is good to compare their result with the Hawaiian plume. As they show, the geochemical zonation is also found for Hawaiian plume, thus, it would be better to introduce this in the section of Introduction. Please also show that the Hawaiian seamount chains only record the plume tail activity, but the present study would shed light on a whole period of mantle plume activity. By showing this it would be more attractive scientifically.

(8) If the authors are sure on the formation age of JCSC, then it should be suspected if the old Tristan geochemical nature can be traced back to 100 Ma. Then it is not clear whether the two-plumelets model was present for the plume head stage?

Guoliang Zhang

Reviewer #3 (Remarks to the Author):

This paper addresses the enduring question of mantle plumes structure and dynamic, which has been extensively debated since more than half a century now. The authors focus on the Tristan-Gough double hotspot track in the South Atlantic that is known to result from a long-lived zoned plume. They provide new isotope data from the less known western arm of the hotspot track on the south American plate (RGR and JCSC volcanic chains). The results confirm that RGR and JCSC are genetically related to the Tristan-Gough zoned plume, providing a more detailed picture of the early history of the plume. Overall, the results support the previously proposed model of two distinct long-lived plume conduits that emplaced following the plume head - plume tail transition.

This paper is definitely an important, high-quality work that deserves publication: it reports high-quality isotope analyses of valuable samples from a remote and poorly known submarine volcanic chain. The results regarding the genetic relation between the western volcanic chains (RGR and JCSC) on the African plate and the Tristan-Gough plume are new and significant. Although speculative, the tentative

reconstruction of the overall dynamic of the plume from the spatial distribution of the Tristan and Gough geochemical signatures on the African and American plates is also a strong point of the paper. The paper is also generally well-written and well-organized.

However, the main conclusions of the paper remain very similar, if not identical, to those of recent papers from the same authors (Rohde et al., 2013; Hoernle et al., 2015; Homrighausen, 2019 and 2020). This is striking when comparing the main conclusions in the abstract to the state of knowledge in the introduction. In my opinion, the paper does not provide new fundamental constraints on plume structure and dynamic. The only new suggestion regarding the stability of the “plumelets” structure through time (“zonation of the Tristan-Gough hotspot can be traced to ~100 Ma ago, thus representing the longest lasting plume zonation recognized thus far on Earth”) is based on tectonic reconstruction and hypothesis on the age of JCSC samples, and not on geochronological data. I expect to find in a Nature Communications paper a new idea or concept/model. I admit this is a difficult objective in the frame of the (very) mature mantle plume topic. For these reasons, I think the paper is better suited to an Earth sciences journal, such as G-Cubed or JGR. Given my recommendation I will not go into a detailed review here.

Response to Reviews

We thank the reviewers for their overall positive assessment of our manuscript and their constructive comments that helped improve the quality of our manuscript. We have carefully revised the manuscript according to the comments and suggestions raised by the reviewers. Please find below a detailed point-by-point response to all comments (reviewers' comments in black, our replies in blue and yellow background color marks text from the article).

Reviewer #1 (Remarks to the Author):

This is a solid paper and contribution to knowledge, with important new observations that contribute to our understanding of mantle plumes. The paper reflects a new field of investigation, which one might call “isotope tectonics” which the authors have pioneered. While the observations of the paper are geochemical--isotopes on samples previously measured for major and trace elements-- the data are not used for geochemical modeling but for constraining mantle flow and surface tectonics. The paper provides new insights into the structure of mantle plumes, and the (surprising) longevity of those structures. The data are important and the data quality is good, the observations are new and important, the conclusions have far-reaching implications. This is a good paper worthy of publication in Nature Communications.

One could argue that the paper is “incremental” since as shown in Figure 5, a long-lived zonation was previously known on the African plate. However, the extension of the data to the Atlantic Plate and the extension in time to 100Ma are important enough in my view that the papers merits Nature communications.

Ordinarily I am a very critical reviewer, as people often use a small amount of data, sometimes of questionable quality, to come to far-reaching conclusions not necessarily well supported by the data. These authors are instead understated, perhaps to a fault. They have a clear question they proposed for a ship-based sampling program—what is the origin of the Rio Grande Rise and how long has the spatial zonation of the Tristan and Gough plume components persisted? And they come up with clear answers. The Rio Grande Rise is not a continental fragment but a manifestation of plume volcanism, and the plume assemblage in the S. Atlantic has produced spatially organized distinct compositions for one hundred million years. It’s amazing, even though the authors adopt a low key approach.

The authors propose a reasonable physical model for the origin of their observations. I suppose that is all that can be required of them, but in my view the paper would have greater impact if the authors considered some of the following--

Homrighausen et al.:

We thank Reviewer #1 for his overall positive assessment of our manuscript and pointing out what he considers to be important points, which were not included in the submitted manuscript, but we have incorporated them to underscore the significance of our manuscript.

To highlight the remarkable result, I would compare the longevity to other known zoned plumes, such as Hawaii and the Galapagos. I think such a comparison would be of great interest to most readers.

Homrighausen et al.:

We thank Reviewer #1 for his suggestion and provide a brief comparison with other hotspots in the section “Spatial geochemical zonation of the Tristan-Gough track” as follows:

“Geochemical zonation is a widespread phenomenon found in numerous plume systems, including the Hawaiian and Galapagos hotspot tracks (e.g., 13, 16, 40). The longevity of the geochemical zonation of the Tristan-Gough hotspot track, however, seems unique. At the Hawaiian Islands (<6 Ma), for example, the Loa and Kea compositional types display two sub-parallel spatial trends (11, 41). The older part of the Hawaiian hotspot track (Emperor Seamounts and Hawaiian Ridge ~85-6 Ma) is dominated by the “depleted” Kea composition without a clear spatial zonation and only sporadic occurrence of the more enriched Loa composition between ~47-6 Ma (42). Spatial zonation of the Galapagos hotspot track, on the other hand, can be traced back to ~20 Ma (13) and possibly was present ~60-90 Ma ago based on data from accreted Galapagos terranes in Central America (40).

In general, hotspot track geochemical zonation has been explained by: (1) an axially-asymmetric, laterally-zoned plume conduit that preserves heterogeneities sampled at the root of the plume through laminar flow (e.g., 12-16, 40, 42, 43), or (2) an unzoned (randomly heterogeneous) plume consisting of enriched fertile lithologies (e.g., pyroxenite/eclogite) within a relatively depleted peridotitic matrix with lateral variations in melting conditions (temperature/pressure; e.g., 9, 10). In the case of the Tristan-Gough hotspot track on the African Plate, variable melting conditions cannot adequately explain geochemical zonation (see Hoernle, et al. (14)). The same is true for the RGR, which shows that the Tristan-type signature is recorded by tholeiitic and alkaline rocks formed under very different melting conditions (32). The long-lived (≥ 100 Ma) and large-scale spatial geochemical zonation along

the Tristan-Gough system can thus be best explained by geographically-separated and geochemically-distinct source compositions.”

Can plumelets really last 100 Ma in a vigorously convecting upper mantle? Since they are relatively low volume, I do not understand why they would not become more dispersed by general mantle flow.

Homrighausen et al.:

We thank Reviewer #1 for this comment and have added the following to our discussion: “This scenario, however, is highly speculative and needs to be confirmed by rheological modelling, as well as the effect of mantle wind, which could lead to motion and/or dispersion of the relatively low-volume plumelets (7); however, local seismic tomography clearly imaged a rising plumelet just southwest of the Tristan island group (28). Additionally, recent buoyancy flux estimates (estimated plume flux divided by two for the proposed plumelets), however, are comparable to the estimates from long-lived plume systems, such as the St. Helena, Meteor/Shona or Louisville hotspot tracks (44).”

Mantle components are good for tectonic mapping such as is done here, but they are really just a description, not an explanation. Perhaps it is discussed in some of their previous papers on this topic, but I wanted to know what was the **cause of the differences between the two discrete EM1 components. I also wanted to know if there were trace element differences between the components that would** explain why there are uranogenic and not thorogenic differences, and what the origin of these components was in the Earth.

Homrighausen et al.:

We have now provided a brief overview of the previously proposed source materials and formation of the distinct EM I-types in a separate paragraph as follows:

“New insights into the origin of Tristan-type EMI and of the geochemical zonation of South Atlantic plumes and their source

As discussed above, the Tristan-type persisted most likely throughout the plume tail stage and is the dominant component at the RGR. Thus, the Tristan-type volcanism is similar in both temporal and areal extent to the Gough-type volcanism in the hotspot track on the African Plate (Fig. 2 & 5). Because the Gough-type is associated with primordial He isotope signatures (see summary in Homrighausen, et al. (20)), the Gough-type is believed to be derived from the base of the lower mantle (14, 15, 20, 45, 46). What, however, is the origin of the Tristan EMI-type composition?

Three distinct EMI-type flavors (Tristan, Gough and southern Discovery) have been identified in the zoned Tristan-Gough, zoned Discovery (northern Discovery has Gough and southern Discovery an even more enriched composition) and Shona (Gough composition) hotspots (Fig. 7a). All samples within one of the three compositional groups form linear arrays on Sr-Nd-Hf and thorogenic Pb isotopic correlation diagrams (e.g., Fig. 3), as do EMI lavas globally (47). The arrays can be explained by mixing of primarily two mantle endmembers: EMI and prevalent mantle (PREMA) or focal zone for oceanic lavas (FOZO; which are equivalent components; 48, 49), perhaps with a minor additional contribution of EMII (enriched mantle two; 49; Fig. 3; 20). The origin of the EMI component is generally related to delaminated ancient metasomatized subcontinental lithospheric mantle (SCLM) and/or lower continental crust (e.g. 17, 25, 47, 50), whereas the PREMA/FOZO components are believed to represent ambient lower mantle (45, 49), possibly containing some EMII representing either subducted marine sediments or upper continental crust (49; Fig. 7b).

In contrast to the other radiogenic isotope systems, the relationship between the two uraniumogenic Pb isotope systems is complex in EMI magma sources, because the half-life for the decay of ^{238}U to ^{206}Pb ($t_{1/2} = 4.47 \text{ Gy}$) and ^{235}U to ^{207}Pb ($t_{1/2} = 0.7 \text{ Gy}$) are significantly different. Thus, most ^{207}Pb was formed during early Earth history (Archean or older), whereas presently most remaining U is ^{238}U , so that primarily ^{206}Pb is formed presently. U/Pb fractionation events happening at different times in Earth history will therefore result in different uraniumogenic Pb growth curves, generating different source compositions. As a result, ancient EMI sources that formed by similar processes at different ages or with different U/Pb ratios can have broadly overlapping Sr-Nd-Hf- ^{208}Pb isotope ratios, but have distinct compositions on the uraniumogenic Pb isotope diagram (47), explaining the difference in $^{207}\text{Pb}/^{204}\text{Pb}$ ratios at a given $^{206}\text{Pb}/^{204}\text{Pb}$ ratio in the different South Atlantic EMI flavors (Tristan lowest, Gough intermediate and South Discovery highest).

Both Tristan-Gough and Discovery hotspots have bilaterally zoned hotspot tracks (Fig. 7a) suggesting that the plumes feeding them are also bilaterally zoned. The most popular explanation for the bilateral zonation is the sampling of different portions of the lowermost mantle with one being from inside the LLSVP (e.g. Gough; Loa at Hawaii) and the other from the ambient mantle surrounding the LLSVP (e.g. Tristan; Kea at Hawaii; 14-16, 42). Since LLSVPs have distinct physical properties compared to the ambient lower mantle (temperature and/or composition; 51) and recent numerical models indicate that bilateral plume zonation requires similar physical properties for both sources (52), both compositional types should either be derived from the ambient mantle or from the LLSVP. Chemical heterogeneity within

one of these reservoirs explains the NNW-SSE oriented geochemical zonation better than sampling the two sides of the N-S oriented LLSVP boundary (inside vs. outside) beneath these hotspots, as previously proposed (Fig. 7a). Interestingly, all 13 oceanic hotspots with EM-type composition are geographically associated with LLSVPs, whereas hot spots located far from LLSVPs exhibit exclusively non-EM compositions (53). The Bouvet hotspot, south of the Shona hotspot, lying outside the projected LLSVP margin (53), has a PREMA/FOZO-type composition with high $^3\text{He}/^4\text{He}$ isotope ratios, and thus most likely represents the ambient lower mantle beneath the South Atlantic (45). In light of these observations, we propose that the different EMI flavors represent delaminated SCLM (\pm lower continental crust) with different ages of U/Pb fractionation, which were stored within the LLSVP until they were sampled by mantle plumes.

There is a broad trend in Atlantic MORB and OIB compositions towards higher $\epsilon_{7/4}$ going from north to south (47). We propose that this large-scale geochemical gradient could ultimately reflect latitudinal compositional gradation of the African LLSVP, resulting from accumulation of delaminated SCLM (\pm lower continental crust) at different times during Earth history. The SCLM could have had progressively different metasomatic ages and/or U/Pb ratios, e.g., reflecting a southward migration of subduction initiation in early Earth history (47). In conclusion, our study suggests that large-scale geochemical grading of the LLSVP, resulting from hemispheric-scale events in early Earth history, can be preserved in complex tree-like plume structures, leading to spatial geochemical hotspot track zonation on the earth's surface (Fig. 7)."

I do not think a reviewer can require those extensions, as each one would involve significant additional work and perhaps additional co-authors such as geodynamicists, but maybe they are worth a paragraph of discussion?

Homrighausen et al.:

We agree with the reviewer that these are important points and therefore have included discussion of them in the paper now. We feel that especially the last addition has also broadened the impact of the paper, by not only explaining the difference in Pb isotopic composition between the different EMI flavors, but also presenting a new model to explain the origin of geochemical zonation in mantle plumes.

Since much of the work in this region has been done by the authors, they are very familiar with it. For the average reader who might be looking at the data for the first time, a **bit more**

background and reporting of previous work would be useful. The authors are to be commended for including the Hoyer et al data in their appendix. Having data on the Tristan and Gough end members would also be useful, particularly since many of the figures have fields for them.

Homrighausen et al.:

The Hoyer et al. data, as well as the data set from the Tristan-Gough seamount chain used for the figures is now added to the appendix. We also thank the reviewer for pointing out these important issues and encouraging us to provide a more detailed background, which we think leads to a significant improvement in understanding the long-term zonation of the Tristan-Gough hotspot.

The statements on lines 414ff that changing the number of samples from n=33 to n=72 might explain the difference in range is to me questionable. You could prove it by taking a random group of 72 samples and boot-strapping 33 at a time and seeing how often range is restricted. Sampling half the population usually does a good job of representing the whole. Maybe instead the difference is real? What then?

Homrighausen et al.:

We thank the reviewer for the comment to the statistical value of our statement. We agree with the reviewer and deleted this point and focused on the temporal evolution to explain the overall lower $^{206}\text{Pb}/^{204}\text{Pb}$ isotope ratios of the RGR compared to the younger Tristan-type rocks on the African plate as follows: “It is worth noting that both the Tristan- and Gough-type lavas evolve to more radiogenic Pb isotopic compositions with decreasing age and that the lowest $^{206}\text{Pb}/^{204}\text{Pb}_{60\text{Ma}}$ ratios (<17.6) were recovered from the oldest (>70 Ma) Gough-type samples (14, 17), whereas previously reported Tristan-type samples (not reaching such low Pb isotope values) are younger. It remains unclear if the temporal evolution to higher $^{206}\text{Pb}/^{204}\text{Pb}_{60\text{Ma}}$ ratios represents a change from plume-ridge interaction (Walvis Ridge) to intraplate volcanism (Guyot Province), or a changing plume source composition with time (17). The RGR samples, which apparently represent the oldest Tristan-type lavas reported so far, appear to extend this trend to even more unradiogenic $^{206}\text{Pb}/^{204}\text{Pb}_{60\text{Ma}}$ ratios. Therefore, we conclude that the RGR and JCSC have a Tristan-type composition and that both Gough- and Tristan-type lavas evolved systematically to more PREMA/FOZO-like compositions through time (17).”

The paper is a solid piece of work that everyone working in this field should read, Even those

peripheral to the field would I think find the results fascinating. With minor revisions I would recommend publication.

Charles Langmuir

Harvard University

Reviewer #2 (Remarks to the Author):

This manuscript reports extensive Sr-Nd-Pb-Hf isotopic data for seafloor volcanic rocks from the Rio Grand Rise and JCSC, and their old ages and Tristan-type isotopic signature lead the authors to suggest a long-lived zonation model for a mantle plume, based on the tectonic reconstruction. I read the paper with interests and find this paper might constitute a good scientific contribution to understanding the nature of mantle plume if the geochemical zonation can be traced back to the plume head stage. I also find some problems that need to be addressed before it can be accepted for publication. I have some comments for the manuscript, and look forward to seeing their reply.

(1) Discussion on the role of alteration: I feel the Section “**the role of alteration**” is not necessary in Discussion. The first paragraph is on the processing procedure and LOI, it would be better to show this in the Results. Moreover, leaching with 6HCl would be sufficient to remove the effect of seawater alteration, this can be verified by a number of papers. The authors should have confidence on their data, thus, it is not necessary to show the correlations between these isotopes, as shown in Lines 247-253, so I suggest to remove this information from the manuscript. The sample description, including alteration characteristics, can be shown in Results.

Homrighausen et al.:

The trace elements were analyzed on rock chips, which were not treated by leaching. We, therefore, believe that a short discussion of the potential role of alteration with the according diagrams is required, especially for the Pb isotope systematics. We, however, follow the reviewer's advice and have moved the alteration paragraph out of the main body of the paper and into the methods section.

(2) For the analytical methods: Leaching with HCl is crucial for Sr-Pb isotopes of seafloor samples, but they indicated only the leaching method for Sr, what about Pb? Leaching with 6NCl at 150C for 3h seems too strong, it would generally be ~70C, I suggest the authors to present the trace element data after leaching if available. They should show not only the measured results of standards but also the reference values, including BCR-2. Presentation of these data are important to understand the data quality.

Homrighausen et al.:

We thank Reviewer#2 for his detailed consideration of our methods and data file. For Nd-Pb-Hf isotope analyses, the sample chips (not powders) were acid-washed in 2N HCl at 70°C for

1 hour to remove surface contamination introduced during hand-picking of the freshest sample material. For Sr isotope analysis, 100-200 mg of rock powder was leached in 6 N HCl at 150°C for 3 hours to remove seawater Sr. We have now stated the procedure as follows in the methods sections “*In order to minimize the effects of alteration and contamination introduced during handpicking of the rock chips for Nd-Pb-Hf isotope analysis, ~100-200 mg of rock chips were acid-washed with 2N HCl at 70°C for 1 hour and subsequently triple-rinsed in 18.2 MΩ H₂O. For Sr isotope analysis, 100-200 mg of rock powder was leached in 6N HCl at 150°C for 3 hours and then triple rinsed in 18.2 MΩ H₂O to remove seawater Sr (with ⁸⁷Sr/⁸⁶Sr=0.7060-0.7090 and Sr concentration of ~8 ppm).*” The trace element data was analyzed on rock powders washed with de-ionized water in an ultra-sonic bath as described by Hoyer et al. (2022). In past studies, we have found no major differences between the parent/daughter ratios of acid-washed chips and powders washed with de-ionized water. Furthermore, the correlation of Th vs Pb and U (Fig.S2) suggests that these elements have not been significantly affected by alteration in our samples.

The BCR-2 standard values are now included in the data file and we apologize that they were not shown in the previous version of the manuscript.

(3) The authors should carefully compare the present study with one of their previous publication Hoyer et al (2022)-CEE. The present paper try to publish the isotopic data separated from Hoyer et al (2022), but the two papers seem to have very similar topics, like continental contamination, tectonic reconstruction, age relationships, et al. To me, Fig. 6 of this ms is quite similar to Fig. 7 of Hoyer et al (2022). I understand the new isotopic data provide robust evidence for the plume model, **but they also need to mention what have already been addressed, and what is new compared to Hoyer et al (2022).**

Homrighausen et al.:

We agree that some topics seem to be similar, such as continental contamination, but in this paper we present Sr-Nd-Pb-Hf isotope data, which in contrast to major and trace elements are not fractionated by differentiation and melting processes and thus serve as a much more powerful tool to evaluate continental crustal contamination. In particular, our main goal is to test if the geochemical zonation can be extended to the RGR. We note that the zonation has only been found in the isotope data thus far and thus cannot be traced with major and trace element data. We have significantly shortened the overlapping topics with Hoyer et al. (2022) and also

added a new section discussing the origin of plume zonation and EMI components in the South Atlantic, which are not discussed in Hoyer et al. (2022).

(4) Section “**Age constraints and implications for the RGR and JCSC lavas**”: the title of this section is mis-leading here. This manuscript does not provide new data of ages, thus, the title should be revised. Also, from my point of view, they should present the information on formation ages in the Section “Geologic Overview”.

Homrighausen et al.:

We thank Reviewer#2 for his suggestion and we now present the important age constraints in the Geological Overview and have removed the section “Age constraints and implication...”. The discussion about the emplacement of the JCSC is now presented in “The spatial geochemical zonation of the Tristan-Gough track”: “The JCSC formation is usually associated with the Eocene volcanism on the WRGR and Cabo Frio at the Brazilian coast (32), located at the NW extension of the JCSC. Multi-beam bathymetric mapping revealed that several sampled volcanic edifices along the JCSC have guyot-like structures (39) and thus can be used to roughly estimate the age since erosion to sea level. Using the depth of the erosional platforms (~1500-2000 mbsl), the estimated age of the underlying oceanic crust (110-90 Ma; Fig. 2b) and assuming normal subsidence of the oceanic crust, we estimate that the JCSC was formed 10-20 Ma after its underlying ocean crust indicating a Late Cretaceous formation. Our plate reconstruction (Fig. 5, & 6) illustrates that both the RGR and JCSC were once located adjacent to the Walvis Ridge and that the spatial geochemical zonation can nearly be traced along the entire combined submarine hotspot track (Fig. 5).”

(5) For Section “**Continental crust embedded in the Rio Grande Rise?**”. I don’t quite understand the logic of this section. As shown in the last paragraph on the previously reported continental rocks in this area should be not be in situ, and thus, does not support the presence of continental fragments under the rise. This information should be at the very beginning of this section. Then they further discuss the possible continental contamination based on isotopic results, but it is not very convincing to exclude the possible origin of EM1 signature from continental contamination. We believe the EMI signature ultimately comes from continental lithosphere, but recycled through the lower mantle and not as a continental silver in the ocean basin. While they propose that Pb-isotopes are good indicator of contamination, then they don’t believe the indication of Pb isotopes on the contamination (Lines 337-340). They prefer to use

MgO and Th/La to preclude the continental contamination. However, MgO is highly dependent on the magma evolution and fractionation processes.

Homrighausen et al.:

We thank Reviewer#2 for his constructive and very helpful suggestions. First, we now focus on the radiogenic isotopes to evaluate the role of crustal assimilation rather than incompatible element ratios that can also be fractionated by melting process, degree of melting, previous melt extraction from a source and differentiation. We, however, note that crustal assimilation will be most pronounced during magma evolution in “shallow” magma chambers. Due to differentiation and potential crustal contamination, we would expect negative linear arrays on MgO vs. $\Delta 7/4$ and $\Delta 8/4$, which is the new figure for this section replacing plots of MgO vs. incompatible element ratios. In accordance with the trace element systematics (Hoyer et al., 2022), the horizontal arrays formed by the WRGR and ERGR on Fig. 4 indicate differentiation without continental contamination. Based on the reviewer’s comments, we have shortened and changed the text in the section as follows:

“The EMI-type flavors of the of the Tristan-Gough seamount chain form distinct fields on the uraniumogenic Pb (Fig. 3a) isotope diagram, whereas the Sr-Nd-Hf- $^{206}\text{Pb}/^{204}\text{Pb}$ - $^{208}\text{Pb}/^{204}\text{Pb}$ isotope ratios largely overlap (Fig. 3b-d,; 14, 15, 20). Our samples have overall lower $^{207}\text{Pb}/^{204}\text{Pb}_{60\text{Ma}}$ ratios at a given $^{206}\text{Pb}/^{204}\text{Pb}_{60\text{Ma}}$ compared to Gough-type EMI and lie within the previously reported Tristan-type compositional array or extend to less radiogenic Pb isotope ratios (Fig. 3a). It is worth noting that both the Tristan- and Gough-type lavas evolve to more radiogenic Pb isotopic compositions with decreasing age and that the lowest $^{206}\text{Pb}/^{204}\text{Pb}_{60\text{Ma}}$ ratios (<17.6) were recovered from the oldest (>70 Ma) Gough-type samples (14, 17), whereas previously reported Tristan-type samples (not reaching such low Pb isotope values) are younger. It remains unclear if the temporal evolution to higher $^{206}\text{Pb}/^{204}\text{Pb}_{60\text{Ma}}$ ratios represents a change from plume-ridge interaction (Walvis Ridge) to intraplate volcanism (Guyot Province), or a changing plume source composition with time (17). The RGR samples, which apparently represent the oldest Tristan-type lavas reported so far, appear to extend this trend to even more unradiogenic $^{206}\text{Pb}/^{204}\text{Pb}_{60\text{Ma}}$ ratios. Therefore, we conclude that the RGR and JCSC have a Tristan-type composition and that both Gough- and Tristan-type lavas evolved systematically to more PREMA/FOZO-like compositions through time (17).”

(6) The authors need to consider a more convincing way to exclude the contamination, for example, they can precisely compare their data with the corresponding crust component on the south America on their figure 3.

Homrighausen et al.:

Based on the study of Santos et al. (2019), the majority of recovered rocks from the RGR has an African affinity and we, therefore, used the lower and upper Namibian crustal composition after Thompson et al. (2007). As recommended, we have indicated the compositional range of the respective components on Fig. 3. Nevertheless, we cannot distinguish between in situ crustal contamination and recycled crust in the magma source with isotope data alone. The lack of evidence for AFC, however, argues against assimilation of in situ crust during differentiation.

(7) Lines 441-452: It is good to **compare their result with the Hawaiian plume**. As they show, the geochemical zonation is also found for Hawaiian plume, thus, it would be better to introduce this in the section of Introduction. Please also show that the Hawaiian seamount chains only record the plume tail activity, but the present study would shed light on a whole period of mantle plume activity. By showing this it would be more attractive scientifically.

Homrighausen et al.:

As also noted by reviewer#1, we now provide a brief comparison of the geochemical zonation along the Tristan-Gough system with the Hawaiian and Galapagos zoned hotspot tracks (see above).

(8) If the authors are sure on the formation age of JCSC, then it should be suspected if the old Tristan geochemical nature can be traced back to 100 Ma. **Then it is not clear whether the two-plumelets model was present for the plume head stage?**

Homrighausen et al.:

We thank the reviewer for this comment and potential misleading presentation of our model. Until now, we have no evidence for a Tristan-type component during the plume head stage. We assume that the two plumelets formed during the transition from the plume head to plume tail stage, which marks an important transition in the temporal geochemical evolution of plume systems. We have provided a potential explanation for this assumption as follows:

“Importantly, the Tristan composition is much more widespread than previously believed and may extend as far back as the breakup of Africa and South America. As noted above, the Sao Paulo plateau could represent the Tristan-type counterpart of the Gough-type Frio Ridge (Fig.

6), extending the geochemical zonation to the opening of the South Atlantic. In this case, the appearance of the Tristan-type composition would coincide with the change from plume head to plume tail stage and indicate that the bifurcated mantle upwelling with two distinct upper mantle plumelets remains relatively stable throughout the entire plume tail stage. The structural change from plume head to tail stage could be caused by a viscosity change between the lower and upper mantle at ~1000 km depth (6, 7). Whereas the massive and buoyant starting plume head was able to punch through this rheological barrier, the subsequent upwelling plume material largely stalled at ~1000 km depth with only smaller plumelets managing to find pathways through this boundary (7).”

Guoliang Zhang

Reviewer #3 (Remarks to the Author):

This paper addresses the enduring question of mantle plumes structure and dynamic, which has been extensively debated since more than half a century now. The authors focus on the Tristan-Gough double hotspot track in the South Atlantic that is known to result from a long-lived zoned plume. They provide new isotope data from the less known western arm of the hotspot track on the south American plate (RGR and JCSC volcanic chains). The results confirm that RGR and JCSC are genetically related to the Tristan-Gough zoned plume, providing a more detailed picture of the early history of the plume. Overall, the results support the previously proposed model of two distinct long-lived plume conduits that emplaced following the plume head - plume tail transition.

This paper is definitely an important, high-quality work that deserves publication: it reports high-quality isotope analyses of valuable samples from a remote and poorly known submarine volcanic chain. The results regarding the genetic relation between the western volcanic chains (RGR and JCSC) on the African plate and the Tristan-Gough plume are new and significant. Although speculative, the tentative reconstruction of the overall dynamic of the plume from the spatial distribution of the Tristan and Gough geochemical signatures on the African and American plates is also a strong point of the paper. The paper is also generally well-written and well-organized.

However, the main conclusions of the paper remain very similar, if not identical, to those of recent papers from the same authors (Rohde et al., 2013; Hoernle et al., 2015; Homrighausen, 2019 and 2020). This is striking when comparing the main conclusions in the abstract to the state of knowledge in the introduction. In my opinion, the paper does not provide new fundamental constraints on plume structure and dynamic. The only new suggestion regarding the stability of the “plumelets” structure through time (“zonation of the Tristan-Gough hotspot can be traced to ~100 Ma ago, thus representing the longest lasting plume zonation recognized thus far on Earth”) is based on tectonic reconstruction and hypothesis on the age of JCSC samples, and not on geochronological data. I expect to find in a Nature Communications paper a new idea or concept/model. I admit this is a difficult objective in the frame of the (very) mature mantle plume topic. For these reasons, I think the paper is better suited to an Earth sciences journal, such as G-Cubed or JGR. Given my recommendation I will not go into a detailed review here.

Homrighausen et al.:

As noted in our manuscript” we believe that our model provides a new perspective of the complex temporal and geochemical evolution of the oldest submarine portion of the Tristan-Gough hotspot track... ”. In this context we also have to mention the overall positive assessment of our manuscript by reviewer#1 and #2, which, for example, wrote: “One could argue that the paper is “incremental” since as shown in Figure 5, a long-lived zonation was previously known on the African plate. However, the extension of the data to the Atlantic Plate and the extension in time to 100Ma are important enough in my view that the papers merits Nature communications.

Ordinarily I am a very critical reviewer, as people often use a small amount of data, sometimes of questionable quality, to come to far-reaching conclusions not necessarily well supported by the data. These authors are instead understated, perhaps to a fault. They have a clear question they proposed for a ship-based sampling program—what is the origin of the Rio Grande Rise and how long has the spatial zonation of the Tristan and Gough plume components persisted? And they come up with clear answers.”

Based on the comments of reviewer 1 and 2, we however have also added a new section to the paper, discussing the origin of plume zonation in South Atlantic hotspots and of the Tristan end member. In this section, we use the longevity of the plume system to point out the stability of the Tristan and Gough source areas and ultimately relate the zonation to delamination of SCLM and storage of this material in the LLSVP. In previous papers the Tristan component was believed to be generated by mixing with MORB (Rohde et al. 2013a) or to sampling lower ambient mantle (e.g. Huang et al., 2011; Weiss et al., 2011; Hoernle et al., 2015; Homrighausen et al., 2019). We go further arguing that plume zonation is derived from a latitudinally compositionally graded LLSVP rather than sampling material on both sides of the margin of the LLSVP and that delamination of ancient SCLM into the LLSVP generated the increase in $^{207}\text{Pb}/^{204}\text{Pb}$ southwards observed in MORB and Atlantic hotspots. This new observation explains why NW to SE chemical zonation is observed in Tristan-Gough and Discovery plumes, rather than E-W zonation expected through sampling of components inside and outside the LLSVP, which would result in the covering of enriched LLSVP lavas by ambient mantle as the plate moves to the NE. We believe that these are major new insights from our study that are distinct from previous studies of zoned plumes globally.

REVIEWERS' COMMENTS

Reviewer #1 (Remarks to the Author):

The authors have been very responsive to reviewer comments and I consider the paper in this revised form would be an excellent contribution and have no further comments.

Reviewer #2 (Remarks to the Author):

Overall, my comments and questions are well, and mostly accurately, addressed. The only flaw to me is lack of geochronological age data of the JCSC volcanic rocks in the paper. The good thing is that the authors have roughly estimated the extrusion age based on geometric and bathymetric results, and the deep (up to 2000 mbsl) erosional platforms implies a rather old age. I would support publication of this manuscript in Nature Communications.

Response to Reviews (second round)

We thank the reviewers for their overall positive assessment of our manuscript and their constructive comments that helped improve the quality of our manuscript.

Reviewer #1 (Remarks to the Author):

The authors have been very responsive to reviewer comments and I consider the paper in this revised form would be an excellent contribution and have no further comments.

Homrighausen et al.:

We thank Reviewer #1 for his overall positive assessment of our revised manuscript.

Reviewer #2 (Remarks to the Author):

Overall, my comments and questions are well, and mostly accurately, addressed. The only flaw to me is lack of geochronological age data of the JCSC volcanic rocks in the paper. The good thing is that the authors have roughly estimated the extrusion age based on geometric and bathymetric results, and the deep (up to 2000 mbsl) erosional platforms implies a rather old age. I would support publication of this manuscript in Nature Communications.

Homrighausen et al.:

We thank Reviewer #1 for his overall positive assessment of our revised manuscript. We have added a figure in the supplementary figure file to illustrate our age estimate with a detailed description. We admit that the preferred age is based on several assumptiony and could only serve as estimate. We also like to mention, that we clearly stated in the manuscript that the “exact emplacement time of the alkaline intraplate volcanism at the WRGR and JCSC remains unclear...”.